

# Sensing earth and environment dynamics by telecommunication fiber-optic sensors: An urban experiment in Pennsylvania USA

Tieyuan Zhu[1,2], Junzhu Shen[1], and Eileen R. Martin[3]

[1]Department of Geosciences, Pennsylvania State University
[2]EMS Energy Institute, Pennsylvania State University
[3]Department of Mathematics, Virginia Tech

**Correspondence:** Tieyuan Zhu (tyzhu@psu.edu)

**Abstract.** Continuous seismic monitoring of the Earth's near surface (top 100 meters), especially with improving the resolution and extent of data both in space and time, would yield more accurate insights about the effect of extreme weather events (e.g. flooding or drought) and climate change on the Earth's surface and subsurface systems. However, continuous long-term seismic monitoring, especially in urban areas, remains challenging. We describe the Fiber-Optic foR Environmental SEnsEing

(FORESEE) project in Pennsylvania, United States, the first continuous monitoring distributed acoustic sensing (DAS) fiber array in the Eastern US. This array is made up of nearly 5 km of pre-existing dark telecommunications fiber underneath the Pennsylvania State University Campus. A major thrust of this experiment is the study of urban geohazard and hydrological systems through near-surface seismic monitoring. Here we detail the FORESEE experiment deployment, instrument calibration, and describe multiple observations of seismic sources in the first year. We calibrate the array by comparison to earthquake data

from a nearby seismometer and to active-source geophone data. We observed a wide variety of seismic signatures in our DAS recordings: natural events (earthquakes and thunderstorms) and anthropogenic events (mining blasts, vehicles, music concerts, and walking steps). Preliminary analysis of these signals suggest DAS has the capability to sense broadband vibrations and discriminate between seismic signatures of different quakes and anthropogenic sources. With the success of collecting one-year of continuous DAS recordings, we conclude that DAS along with telecommunication fiber will potentially serve the purpose

of continuous near-surface seismic monitoring in populated areas.

## 1  Introduction

As increasingly more people reside in urban areas, and as climate change leads to more extreme weather patterns, we need more reliable data to understand the Earth's surface and subsurface systems and design appropriate strategies to estimate risks

and reduce the vulnerability of people in cities. The upper 100 meters of the subsurface, also called the critical zone or near-surface, is a major driving factor behind geological and hydrological systems' behaviors and is the layer that most affects stability of infrastructure and buildings. However, near-surface Earth materials driven by multiscale physical, chemical, and



biological processes are extremely heterogeneous, varying spatially at the scale of meters or even smaller and temporally from

milliseconds (or less) to millions of years. This suggests that dense and continuous measurements that can yield spatiotemporal information are particularly valuable, and such data could be provided by a dense and (semi)permanent deployable seismic array.

Despite their expected utility for real-time monitoring of subsurface environmental systems, neither temporary nor permanent dense seismic geophone arrays have been widely deployed in urban areas. The primary limiting factors in deploying traditional seismic monitoring include: restrictions on human-controlled sources in densely populated areas, difficulty obtain-

ing permission and space to deploy sensors near civil infrastructure, challenges in securing sensors against theft or vandalism, and high costs to maintain power and data transfer from geophones.

While traditional seismic monitoring is infeasible in urban areas, a rapidly developing technology, fiber optic distributed acoustic sensing (DAS), provides a promising alternative. The DAS technique repurposes a standard fiber-optic cable with an attached laser interrogator unit (IU) as an array of dense pseudo-seismometers measuring vibrations by repeatedly probing the

axial strain rate at all points along the entire fiber (Parker et al., 2014). DAS was initially applied to geophysical applications to record active seismic sources in rural or offshore areas for oil and gas exploration and monitoring, as well as seismic monitoring of $CO_2$ sequestration (Daley et al., 2013). Later, DAS was applied for near-surface shear wave imaging (Dou et al., 2017), permafrost thaw monitoring (Ajo-Franklin et al., 2017; Martin et al., 2016), and monitoring water table levels (Ajo-Franklin et al., 2019).

Several experiments in California have been carried out by teams using existing telecommunications infrastructure. By using existing telecommunications infrastructure, particularly by plugging into "dark" or unused fiber that is already installed underground, these experiments greatly reduce the experimental cost and setup time as an interrogator simply needs to be plugged into one end of a stretch of fiber to being data acquisition. This series of experiments have shown that signal quality is often good enough for earthquake detection and imaging even though loose cables in underground conduits only couple to the

surrounding soils through friction and gravity (Lindsey et al., 2017; Jousset et al., 2018; Martin et al., 2019; Yu et al., 2019; Ajo-Franklin et al., 2019). At the Stanford Fiber Optic Seismic Observatory, Rayleigh wave dispersion showed significant spatial variability at scales relevant to earthquake ground motion prediction (Martin, 2018; Spica et al., 2020), and time-lapse changes through a building excavation (Fang et al., 2020). However, investigation of near-surface time-lapse changes due to seasonal precipitation variation yielded no significant velocity variation when investigated with time-lapse ambient noise interferometry

(Martin, 2018).

In this study we introduce the Penn State Fiber Optic foR Environmental SEnsEing (FORESEE) project. The ultimate goal of this project is to understand the response of DAS fiber sensing arrays to particular events and use repeating signals to continuously monitoring environment and subsurface physical/chemical/biological changes. Similar to the Stanford array (Martin et al., 2019) and recent Pasadena array (Zhan, 2020), the Penn State FORESEE array continuously records DAS data

along 5 km of dark underground telecommunication fibers for over one year since April 2019 (Zhu and Stensrud, 2019). This is the first deployment of a DAS dark fiber array in the Eastern US.





This new experiment will improve our understanding of the reproducibility of results across varied installation types, but will also provide several new opportunities. Firstly, soil and shallow bedrock in the Allegheny Mountains region creates complex near-surface geophysical properties with strong heterogeneity (Brantley et al., 2013). Secondly, strong seasonal variations
in temperature and precipitation yields a unique opportunity for understanding the sensitivity of DAS to temperature and groundwater level fluctuation. In addition, karst geology systems in most of Pennsylvania evolve through hydrologic processes. The underlying carbonate bedrock can be slowly dissolved by circulating groundwater, which can form sinkholes and caverns causing potential hazards on relatively short geologic time scales (Bansah, 2018). Especially in urban areas, sinkhole collapse and subsidence issues can be extreme threats to human safety and property (Weary, 2015). In terms of earthquake research,
although the Eastern US typically experiences less seismicity, the underlying bedrock is older, harder and often denser than in the Western US, which allows seismic waves to propagate more efficiently in the event of an earthquake. In addition, eastern cities are geographically dense and have many older structures built before the 1970s, which were not designed to endure earthquakes. More extensive sensor systems will benefit the characterization of regional earthquake hazards.

In this paper, we detail the field deployment of the FORESEE array. Then we report a variety of interesting signals, including
global and regional earthquakes, thunderquakes (Zhu and Stensrud, 2019), and mining blasts. Further, we show some surprising anthropogenic noise recordings: footsteps, and live music. We conclude with discussion of these data and the important role they may play in understanding the subsurface and infrastructure.

## 2 Experiment Overview

In designing the array, we had several goals: high resolution to detect small-scale subsurface features, long aperture to increase
coverage of the area and increase likelihood of sensing near a feature of interest, ease of access following existing telecommunications fiber paths, and including at least two directions. Considering these aims, we selected the fiber route pictured in Figure 1a consisting of two fiber optic sections spliced together (around channel 1340), a total fiber length of approximately 5 km. These fibers are all underneath the Pennsylvania State University campus, and this experiment used a single strand of fiber-optic in each cable. These cables were already being used for telecommunications purposes, but not all strands were
previously in use (so-called "dark fiber"). These cables were sitting in buried concrete conduits at a depth of roughly 1 meter, as pictured in Figure 1b.

The DAS measurements are recorded by a Silixa iDAS2 interrogator unit, pictured in Figure 1a, which is connected to one end of the fiber using an E2000 APC connector. The DAS array made continuous strain rate measurements at a 500 Hz sampling frequency with a 10 m gauge length and 2 m channel spacing. We began recording data on 5 April 2019 and the
recording is still running at the paper submission.

### 2.1 Data storage

Because this experiment generated many tens of terabytes of data, we connected the DAS interrogator unit to a network attached storage (NAS) server. The server was connected to an internet network providing us with remote data access in real-time. The





interrogator and NAS were hosted in a Penn State IT building connected with an uninterruptible power supply (UPS) for backup

power in case of station power interruption. Figure 1 (top-left panel) shows the setup of the DAS system of the FORESEE pilot experiment in a standard computer rack. A GPS clock and antenna were connected for precise timing. We installed the GPS antenna outside of the building, roughly 2.0 meters above the ground. Raw DAS recordings are saved in TDMS format (Silixa iDAS default data format, for which Silixa provides Matlab and Python reader scripts). The selected DAS recording settings yielded over 200 GB/day and ultimately about 76 TB raw data per year.

## 2.2 Determination of sensor locations

Before any seismic array processing, we needed to determine the precise location of each sensor along the fiber. Because the fiber sits in underground conduits, it is invisible from the surface, so traditional methods of obtaining GPS coordinates at individual surface seismic sensors were not feasible. Penn State Enterprise Networking  Communication Services department manages the cable, so they provided a map of the fiber cable path and telecommunication manholes visible at the surface.

Following this map, we ran tap tests to determine a set of representative channel indices at particular locations. For each location, we used a hammer drop source to generate six distinguished shots on the ground. Ideally the channel centered around a shot will respond most strongly to the shot, and correspond to the GPS position of the hammer source. At many locations, strong noise inhibited our ability to identify hammer shots in raw DAS data. Instead, we compute the spectral energy, where is the Fourier Transform of windowed by a short sliding window (Martin, 2018; Zhu and Stensrud, 2019). As seen in Figure 5b,

the hammer shots are easily identified and the center channel is relatively unambiguous. We refer readers to our previous work for examples (Zhu and Stensrud, 2019). We ran more than 101 tap tests to obtain the positions of 101 channels. Finally we used interpolation to estimate locations of all 2137 channels with 2 m spacing. Note that some channels were removed from analysis after determining locations where fiber was looped back on itself.

## 2.3 DAS calibration to seismometers

To verify that this DAS system (interrogator, fiber, and cable in conduit) records a useful metric of ground motion, we convert our DAS recordings to particle velocity and compare to the nearest broadband seismometer. This calibration procedure is based on several previous studies (Daley et al., 2016; Wang et al., 2018; Lindsey et al., 2020a). To ensure this paper is self-contained, we briefly review this procedure.

The interrogator that we used records axial strain rate in the direction of the cable. Specifically, the actual output of DAS system is the phase changes between consecutive pulses during the laser repetition rate (Figure 2). The phase of light traveling a distance $x$ in a cable with refractive index $n$ can be expressed as

$$\Phi = \frac{4\pi n x}{\lambda}$$

. The wavelength of Rayleigh backscattered light equals the incident wavelength (1500 nm). We assume refractive index

changes linearly with strain rate ($\frac{\Delta n}{n} = (\zeta - 1)\frac{\Delta x}{x}$). The scalar multiplicative factor ($\zeta$) is determined by the material properties: $\zeta = 0.735$ for single-mode fiber glass with light propagating inside (Lindsey et al., 2020a). Hence the phase changes at the same





fiber section separated by the gauge length $L_G(L_G = 10m)$ can be expanded as:

$$\Delta\Phi = \frac{\partial\Phi}{\partial x}x + \frac{\partial\Phi}{\partial n}n = \frac{4\pi nx}{\lambda x}\Delta x + \frac{4\pi nx}{\lambda n}\Delta n = \frac{4\pi nL_G}{\lambda}\left(\frac{\Delta x}{x} + \frac{\Delta n}{n}\right) = \frac{4\pi n\zeta L_G\Delta x}{\lambda x} \tag{1}$$

After dividing both side with sampling interval $\Delta t$ and rearranging the equation above, we can link the DAS output to strain
rate $\dot\epsilon_{xx}$ by applying a constant scaling factor as follows

$$\dot\epsilon_{xx} = \frac{\Delta x}{x\Delta t} = \frac{\lambda}{4\pi n\zeta L_G\Delta t}\Delta\Phi = \frac{1500\times10^{-9}(m)}{4\pi\times1.445\times10(m)\times0.735}f_s\Delta\Phi = 11.6\times10^{-9}f_s\Delta\Phi \tag{2}$$

where $f_s$ is the sampling rate. In our case, $f_s = 500$ Hz.

To illustrate the calibration procedure, we take Peru M 8.0 earthquake in May 26 2019 as an example to convert DAS
recordings to particle velocity. We chose over 80 channels of a L-shape subarray with 40 channels in each direction, shown in
Figure 4. Traces from the DAS cable segment of the same orientation are compared with one horizontal component rotated to
the cable direction of nearby seismic station (SSPA) 17.7 km away. This distance would be too far for a comparison in response
to a local seismic source, but this teleseismic event is expected to yield similar responses by seismometers at this distance. The
steps to this comparison are outlined in Figure 3. First we multiply the raw data by the constant scaling factor $(11.6\times10^{-9}f_s)$ to
strain rate (nanostrain/s). The strain rate is integrated along time axis to obtain the strain. Low frequency artifacts resulting from
integration are removed by applying a bandpass filter between 0.02 and 0.5 Hz. Then DAS array strain values are converted to
particle velocity values ($\mu m/s$) in the frequency-wavenumber (FK) domain using $v = \frac{\partial u}{\partial t} = -c\frac{\partial u}{\partial x} = -\frac{\omega}{k}\frac{\partial u}{\partial x} = -\frac{\omega}{k}\epsilon_{xx}$ (Daley
et al., 2016). In the code, we apply the f-k transform of the seismograms with 40 traces and rescale Fourier coefficients by
$V(\omega, k) = -\frac{(\omega+\sigma)E(\omega,k)}{(k+\sigma)}$, where $\sigma$ is the threshold to avoid instabilities in division by zero frequency and wavenumber values
(Lindsey et al., 2020a). Finally we apply inverse Fourier transform to $V(\omega, k)$ to obtain particle velocity in the time domain.

Figure 4 shows the comparisons of DAS recordings at two orthogonal directions (Figure 4a) to reference seismograms
(BH1 and BH2) (particle velocity in $\mu m/s$) at the nearest seismic station (SSPA) after applying a bandpass filter $0.05 - 1.0$
Hz. It is clear that the DAS data shows an agreement with the reference seismogram (particularly S wave in BH2). Small
inconsistencies in coda wave amplitudes are possibly due to the DAS data represents the average signals over a gauge length
rather than as an isolated point sensor (Wang et al., 2018), although many other factors may play a role in waveform differences,
e.g., the different locations of the seismic station and DAS array, directional sensitivity, and the host environment. Overall, this
calibration process suggests that this DAS array is an acceptable system for recording single component low frequency seismic
data.

To calibrate higher frequencies, we performed a similar comparison using an active source, shown in Figure 5. We co-located
a 24-channel geophone array (4 m spacing) on the ground just above the fiber cable. After synchronizing the GPS timing, we
searched for DAS shot gather recordings. We applied a bandpass filter 10-60 Hz and auto-gain control (AGC) to geophone and
DAS data. In the geophone data, we identified a direct wave (∼800 m/s) and refraction (∼4000 m/s). From this we inferred
that the depth of the first layer is approximately 8 m. In DAS, there is more coherence in the moveout of the waveform, and
it appears that the direct wave is about 2250 m/s. We hypothesize that this seems to be the average of the direct arrival and
refraction on the geophone gather. We identified four possible reflections (red arrows) that are kinematically consistent in both





the geophone and DAS data in Figure 5. Dynamically, these phases in DAS are more continuous and coherent, possibly owing to the continuous nature of the overlapping DAS channels, compared to discrete geophones.

Ground roll (surface waves ∼370 m/s) is clear in the geophone gather but not manifested in DAS. Results on this have been mixed in different experiments. Martin et al. (2017b) conducted an active-source seismic experiment with fiber in underground telecommunication conduits and did not report strong ground roll that was clear on 3C nodes. However, Spikes et al. (2019)

conducted an active-source seismic experiment to compare geophone and DAS data where the fiber was left on the surface of the ground. Their data showed clear surface wave moveout in DAS. There are several possibly hypotheses behind the lack of surface wave in Figure 5b, e.g., fiber buried at shallow depths (∼1 m) in the conduit and loose contact of fiber to the conduit.

Another unique and interesting observation on DAS is the energy around the hammer shot shifted down (highlighted yellow zone), which is likely the trapped waves propagated vertically inside a low velocity zone (e.g., sinkhole in this area). This

observation echoes the study of fault-zone trapped waves using dense seismic geophones and DAS by Jousset et al. (2018). The ability to detect such signals is a benefit of using dense sensors like DAS. We suggest that further seismic wave modeling should be conducted to verify the presence of the low velocity zone.

## 3   Ground Motion Induced by Natural Sources

Several prior studies have demonstrated that earthquake signals can be recorded by DAS with newly installed fiber (Lindsey

et al., 2017; Biondi et al., 2017; Wang et al., 2018) and dark fiber in the conduit (Lindsey et al., 2017; Ajo-Franklin et al., 2019; Yu et al., 2019). Here we report observations of earthquakes and thunderquakes from Penn State FORESEE array. We further include a comparison of the array to standard seismometers at both low and high frequencies, a necessary step to verify the broadband response of any new DAS array (Lindsey et al., 2020a).

### 3.1   Earthquake

Figure 6 shows DAS recordings of four earthquakes (Peru M8.0 earthquake on May 26 2019, Ridgcrest M7.1 earthquake on July 06 2019, Tennessee M3.8 earthquake on January 20 2020, and PA M1.1 earthquake on August 27 2019. We applied the bandpass filtering to both PSRS and DAS seismograms. All top panels show seismograms (particle velocity) from nearby seismic station PSRS as references and one trace (100th sensor) from DAS while the bottom in Figure 6 shows full DAS recordings of 2137 channels. DAS of Peru M8.0 earhquake shows strong P-wave, S-wave, and surface waves. It serves well

to calibrate kinematics and dynamics of seismograms in the above section. While, for Ridgcrest M7.1 earthquake, P-wave are relative invisible in DAS, S-waves are clearly identified in 03:27:30, being consistent to seismometer S-wave. And DAS surface wave energy is much stronger than S-wave. The Tennessee M3.8 earthquake appears in weak energy in DAS recordings in Figure 6c. Due to rush hours (3:10 PM local time), this DAS record is very noisy and contaminated with traffic noises (linear events), but surface wave is still identifiable. Last event, shown in 6d, was the PA M1.1 earthquake nearby the State College

(10 km away from the array) in Pennsylvania. Two phases (probably body and surface waves) are clearly visible.



## 3.2 Thunderquake

While there have been other examples of DAS arrays using existing telecommunications infrastructure on land, these have been concentrated in the Western US, which rarely experiences thunderstorm lightning (Changnon, 2001). Thus, in deploying an array in the Eastern US, we had a unique opportunity to study how thunder and lightning couple to the ground to induce seismic waves. Prior studies with single or small-handfull-of seismometers have observed that thunder can induce ground motion (Lin and Langston, 2007), but to our knowledge this has never been studied with a large, dense array (Zhu and Stensrud, 2019). On April 15 2019, a severe thunderstorm crossed over the array, verified by the National Lightning Detection Network (NLDN). We were able to detect clear thunder events across the array, one of which occurred in April 15 2019 is pictured in Figure 7 (Zhu and Stensrud, 2019). Figure 7a shows 2-min raw recordings of six (e4-e10) events between UTC 03:33 - 03:35. Spectrograms of two channel traces (black traces overlain in Figure 7a) show the spatial variation of recordings between two channels. We hypothesize that these recorded seismic energy is induced by thunder and/or lightning electromagnetic waves coupling to the ground to induce surface waves propagating in the shallow subsurface. The spatial variation could suggest the spatial attenuation of the shallow subsurface layer. The picked traveltime moveout enable us to characterize the thunderquake events (Zhu and Stensrud, 2019) and wave propagation across the FORESEE array even State College can also be reconstructed from DAS data. Until now, we already manually identified more than 120 thunderquakes from other four severe thunderstorms from April to August 2019.

Figure 8 shows the average power density spectrum of raw DAS recordings of the earthquake and thunderquake events in Figures 6 and 7, which is computed by averaging the power spectrum of each channel for all channels. The Nyquist frequency of DAS recordings is 250 Hz. Distant earthquakes were downsampled in preprocessing. We observed low frequency local and regional earthquakes (PA M1.1 and TE M3.3) [0.05 − 20 Hz] and a long-period global earthquake [0.01 − 1 Hz]. The peak of thunderquake lies in the range of high frequency [10 − 130 Hz]. We can conclude that the FORESEE DAS fiber is able to record the broadband quakes [0.001 − 250 Hz].

## 4 Effect of Geometry on Measurement Polarity

We found that in many cases channels oriented in orthogonal directions show a polarity flip following the S waves. Figure 9 shows an example of this polarity flip from Peru M 8.0 earthquake (Figure 6a). The recording of the seismic waves from far-field, which can be considered as plane waves, should be constant over the entire array. However, after the arrivals of S waves, two straight sections of fibers in different directions at the corner (see BH1 and BH2 in Figure 4a) show the waveforms with opposite signs, while the polarity in the window of P waves remains unchanged. This pattern is caused by the axial sensitivity of DAS and the tensor nature of strain measurements, and this observation is in agreement with prior observations and theoretical modeling (Lindsey et al., 2017; Martin, 2018; Martin et al., 2018b). DAS only records the strain rate along the fiber and has different response of P and S waves since they have different polarization. For incoming waves with particle motion in the same direction as propagation (e.g. P waves, Rayleigh waves), the recordings of two orthogonal fibers have the same polarity but the amplitude is determined by the wave propagation direction. For waves with particle motion perpendicular to the fiber direction




(e.g. S waves, Love waves), the amplitude is the same but the polarity flips (Lindsey et al., 2017). This understanding of the

polarity and amplitude effects of geometry allows us to approximately predict the response of the fiber in different directions to some known wave mode arrivals.

## 5  Urban Anthropogenic Seismic Sources

The high degree of human activities in urban environments results in a large level of background vibrations that has been received scientific interests in terms of subsurface characterization and hazards mapping (Díaz et al., 2017). Prior urban DAS

studies have shown anthropogenic sources of noise such as pumping systems (plumbing, heating or air conditioning) and vehicle traffic (Martin et al., 2018a). While vehicle moving signals can be identified by the linear moveout events and the passing speed is calculated by the slope of the signal, the section will present the DAS recordings of other interesting anthropogenic sources not previously reported.

### 5.1  Footsteps

It is surprising that despite relying on just friction and gravity for coupling existing fiber optics to conduits, we were able to detect walking individuals in Figure 10. After bandpass filtering [$1 - 5$ Hz], clear footstep signals are recorded by a subset of array beneath a straight path without moving cars. With offsets in this dense array in the bottom panel of Figure 10, we are able to identify the walking direction and whether vibrations are caused by individuals or groups (see examples of individuals and groups in Figure 10); the second author (black dashed-line) following a group of people (blue line) departed from the

bus station (around channel 1270) and headed south; and later a walker (red line) moved in the opposite direction towards the bus station, corresponding well with the timings of in-person field observations made on July 15 2019 (Figure 10). We can also estimate the walking speed about 1.2 m/s from the slop of dashed lines. To be general, we analyze the spectral analysis of one-minute data at noon (across channels 800-2137, above central campus) from three very different days (April 16 2019, February 16 2020, and April 16 2020). Figure 11 shows significant peaks at around 2.0 Hz and 4.0 Hz (harmonic), while they

disappear at midnight. This 2.0 Hz signal corresponds to walking steps on campus, equivalent to 120 steps per minute, which is similar to the people walking on shopping floors with an average frequency of 2.0 Hz and a velocity of 1.4 m/s (Pachi and Ji, 2005), slower than marathon runners with 2.8 Hz (Diaz et al., 2020). Not surprisingly, this footstep signals are not shown up on April 16 2020 during the COVID-19 crisis after the stay-at-home order in PA on April 1 2020. And its noise power spectra level is close to that at midnight. Spatially, we also calculated the power spectra of data at channels 1-600 (on the edge

of campus) and the peaks at 2 and 4 Hz are not found (not shown here). We can see that DAS recordings provide the spatial and temporal distribution of the footsteps that may be useful for designing and analysing campus traffic.

For the goals of the FORESEE array, these footsteps are likely to hinder efforts towards subsurface imaging with ambient noise interferometry, so we are interested in removing these signals prior to imaging. Further, as DAS arrays are deployed in a wider variety of locations, there may be areas where removing the footsteps is needed to ensure privacy of people in the area.





We recently developed a convolutional neural network to automatically identify pedestrian footsteps, the first step towards removing these signals (Jakkampudi et al., 2020).

## 5.2    Mining blast

Several mining sites around the State College (Figure 12) provide well-repeatable blast sources for further calibration and near-surface monitoring. The distances of four mining sites (site 1, site 2, site 3, and site 4) to the campus are about 31 km, 41

km, 84 km, and 15 km, respectively. Reference seismic station indicated by the blue triangle is about 10 km west-south away from State College. Figure 14 shows three quarry blast events from site 1, site 2, and site 3, respectively. With proper bandpass filtering, these events are clearly visible in the noisy records since they were occurring in traffic hours. Their recordings are kinematically consistent with reference seismograms. Strong surface waves are identified despite of these explosive sources. In 2019 there are more than hundreds repeatable blast events cataloged from these sites, which could allow yearly near-surface

monitoring of geotechnical engineering activities (Fang et al., 2020) and/or hydrological systems owing to a significant ground water level variation. One strong event, shown in Figure 14, was recorded on May 14 2019 from the site 4 (Pleasant Gap, PA). Since this event has equal distance seismic station PSRS and DAS (see Figure 12), there is a 2-sec time delay between two data. High frequency body- and surface waves (10-20 Hz) are identified. Same as previous observations, low frequency surface waves exhibit the flipped polarity in the orthogonal fiber locations (e.g., indicated by arrows in channels 170 and 600).

## 5.3   Live music

In general, seismic sources that only couple to the ground through sound often have weaker coupling, and prior studies of active seismic source experiments detected by buried fiber optics did not show air waves (Martin et al., 2017b). However, our DAS data recorded distinctive signals corresponding to live music during the April 26 2019 Penn State Movin'On music festival (Zhu et al., 2019). The live music stage was directly above channel 120 - 150. Figure 15 shows an example of 20-min

DAS recordings of four songs "Welcome to Your Life", "Cannonball", "Good Morning", "Ways to Go" based on the timings of the concert playlist. Figure 15b shows large variations of recorded seismic amplitudes, even within the different parts of a single song. The break between songs is easily identified as a gap between waveforms and spectrograms in Figure 15c. The spectrogram of trace 130 shows that different song results in a characteristic spectra composed by narrow and evenly spaced energy (zoomed details in Figure 15d). We can detect sustained notes in the bass range (40-140 Hz). Played back and visualized

with the IRIS SeisSound tool, these signals are clearly fat bass rifs with much of the energy below 100 Hz (also audio of "Good Morning" song in supporting materials (Zhu, 2020)). This is not the first time seismometers have detected the bass line of concerts; Díaz et al. (2017) showed similar results from a single broadband seismic station during a Bruce Springsteen concert in Barcelona. The difference here is the densely sampled spatial data, which enables us to see that these signals were clearly sensed more than half a kilometer away in Figure 16. Wang et al. (2020) reported similar seismic recordings of parade floats

and bands by the Pasadena distributed acoustic sensing array.



## 6   Discussion

This experiment adds a unique geological environment and new set of questions to the growing body of research on the use of DAS in populated areas and around infrastructure. While these new sensing systems have some limitations, they also have a number of benefits, and are enabling a wider variety of applications in new locations. In particular we are seeing their value

in the Eastern US, and our investigations suggest DAS systems could play a significant role in the broader ecosystem of smart city development.

A primary limitation of DAS arrays at present is that each channel records the axial strain rate. On a straight fiber optic cable this is a single component of the strain tensor. While some fit-for-purpose installations for energy production or $CO_2$ sequestration have utilized helical fibers to instead record a mixture of strain components (Kuvshinov, 2016), the cost of producing

these specialty cables is typically too high for engineering and environmental geophysics. Thus, dark fiber arrays must find creative methods to utilize different directions within an array made up of straight segments. Dark fiber DAS acquisitions have the additional limitation that we currently have only a small understanding of the effects of the installation on signals (Papp et al., 2017; Martin et al., 2017a; Ajo-Franklin et al., 2019).

Despite those limitations, DAS technology has enabled dense wide-aperture sensor arrays with very little labor approaching

industry-scale exploration. In particular, the density of sensors has enabled a wider variety of methods to image subsurface structures and properties, including full waveform migration and inversion (Egorov et al., 2018) and receiver function Moho imaging (Yu et al., 2019). For near-surface imaging, Zhang et al. (2020) successfully applied wave-equation dispersion inversion to ambient noise DAS data with careful processing. At the FORESEE array, the combination of wide aperture and high density enabled full waveform modeling and time-reversal imaging to characterize thunderquake sources as a new source of

strong local seismic energy (Zhu and Stensrud, 2019).

In applications where seismic energy sources of interest are spread over wide areas, particularly when using earthquakes or thunderquakes as sources for imaging, fiber optics will enable extensive coverage. This is particularly important in the Eastern US, which has relatively low rates of seismicity, meaning fewer regional earthquakes and less traditional instrument coverage to capture high frequency content. This leads to limitations in high-resolution regional and urban near-surface models, so other

seismic energy sources such as thunderquakes could be particularly useful in developing 3D regional tomography maps of the shallow crust beneath local regions in the Eastern US.

Moving forward, DAS arrays utilizing existing telecommunications fibers are making it much more cost-effective and practical in urban areas than installing traditional instrument arrays, and DAS arrays can play an increasing role in development of resilient, sustainable cities. This includes geophysical and geotechnical applications: near-surface imaging for planning sta-

ble structures, measuring ground motion due to natural sources, monitoring subsidence, identifying major sources of seismic energy, understanding urban hydrological systems, and locating geohazards. The value of DAS has been recognized in inaccessible and harsh environments, enabling offshore ocean observations (Lindsey et al., 2019; Williams et al., 2019), as well as Arctic monitoring as climate change threatens the stability of permafrost under infrastructure (Martin et al., 2016; Ajo-Franklin et al., 2017) and leads to degradation of glaciers (Walter et al., 2020). We anticipate that it will also play an important role in



the critical zone community to image near-surface heterogeneous Earth materials, varying spatially at the scale of meters or even smaller and temporally from hours to years. Further, unprecedented large-volume DAS data provides an opportunity to test new data analytic algorithms.

However, the economics of deploying fiber optic systems are unlikely to be motivated by geoscience alone, and we must understand DAS arrays as multipurpose systems with a variety of applications in engineering and urban planning. In general,
detection and identification of small events in the noisy urban environment has been challenging, but we can take advantage of the dense and continuous recordings provided by DAS to isolate these noises to understand and remove them. In particular, both unsupervised and supervised machine learning methods have been used to isolate car signals and footsteps for removal (Martin et al., 2018a; Jakkampudi et al., 2020). Car detection can even yield insights into temporally and spatially varying patterns of human activities relevant to public health and urban planning (Lindsey et al., 2020b). Additional future applications
beyond geophysics should be studied, including traffic monitoring and redirection (without requiring private cell phone data), gunshot array detection, industrial noise pollution monitoring, and subsurface water utility monitoring.

## 7 Conclusions

We have deployed the FORESEE array using existing fiber optics under the Penn State University Campus in Pennsylvania, USA, and acquired 75 TB of data over the course of about 360 days since April 05 2019. While this array confirms findings from
earlier dark fiber arrays in the Western US that such a system can record local active seismic sources and earthquakes, this is the first experiment of its kind in the Eastern US, and reveals a wider range of new signals including thunderquakes, concerts, and even footsteps. The density of these broadband DAS recordings provide extraordinary resolution that enables insight into their cause and allows us to distinguish between these various signals. We anticipate that the collected FORESEE data will be able to answer relevant geoscience questions particularly related to urban hydrology and geohazards. DAS arrays utilizing existing
telecommunications fibers have the capability to sense broadband vibrations, and we conclude that DAS will potentially serve as a multi-purpose system for continuous near-surface seismic monitoring in populated areas (e.g., geohazards, critical zone, permafrost, hydrology, geotechnical engineering, infrastructure management and urban planning).

*Data availability.* The Penn State FORESEE DAS processed waveform data will be available via Penn State Data Commons. The down-sampled thunderquake data is available here: https://sites.psu.edu/tzhu/foresee/. Broadband seismic waveform data for the PSRS station are
retrieved from the IRIS Data Management Center (doi.org/10.7914/SN/PE). Figure 1 map is available under the Open Database Licence.

*Video supplement.* Waveform and audio of 3-min DAS live music signals during 21:02 – 21:05 UTC on April 26 2019.



*Author contributions.* TZ designed the experiment, conducted data processing and analysis, and led the writing. JS managed DAS data, calibrated DAS data, and contributed to data analysis and the writing. ERM assisted in experiment planning, contributed to data analysis and the writing. All authors participated the field work.

*Competing interests.* There is no competing interests.

*Acknowledgements.* We really appreciate Chris Marone for his warm support to convince the Penn State Institute of Natural Gas Research to provide seed money to the FORESEE array. We would like to thank our collaborators Patrick Fox, Dave Stensrud, and Andy Nyblade for their contribution of the FORESEE array. We thank Todd Myers, Ken Miller at Penn State University and Thomas Coleman from Silixa who help setup the fiber-optic DAS array. The Penn State FORESEE array was supported by Penn State Institute of Environment and
Energy seed grant and Institute of Natural Gas Research. E.R.Martin was supported by DOE Award No.DE-SC0019630 and by DOE Award No.DE-FOA-0001990.





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





(a)



(b)

**Figure 1.** (a) Penn State fiber-optic distributed acoustic sensing (DAS) array map. Numbers listed along the array denote the channel number. Top-left: DAS field setup; bottom-left: a photo of tap tests; bottom-right: the fiber end. (b) Cartoon of DAS array connecting existing fiber optics. The left-bottom subfigure shows the pre-existing fiber-optic cable (in black) used for this project. Map @OpenStreetMap contributors 2020. Distributed under a Creative Commons BY-SA License.



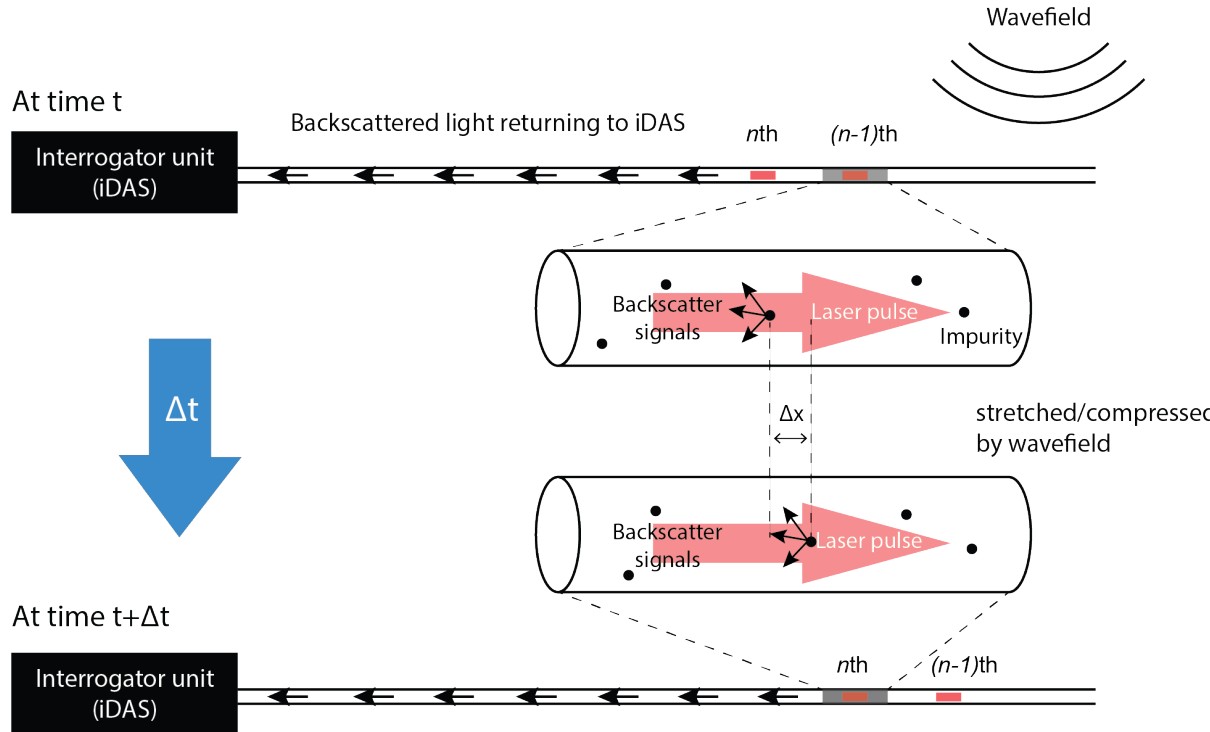

**Figure 2.** Principle of DAS. Rayleigh backscattering occurs at anomalies in the optical fiber which can be shifted by the surrounding wavefield. The phase changes of successive pulses are recorded by iDAS instrument.

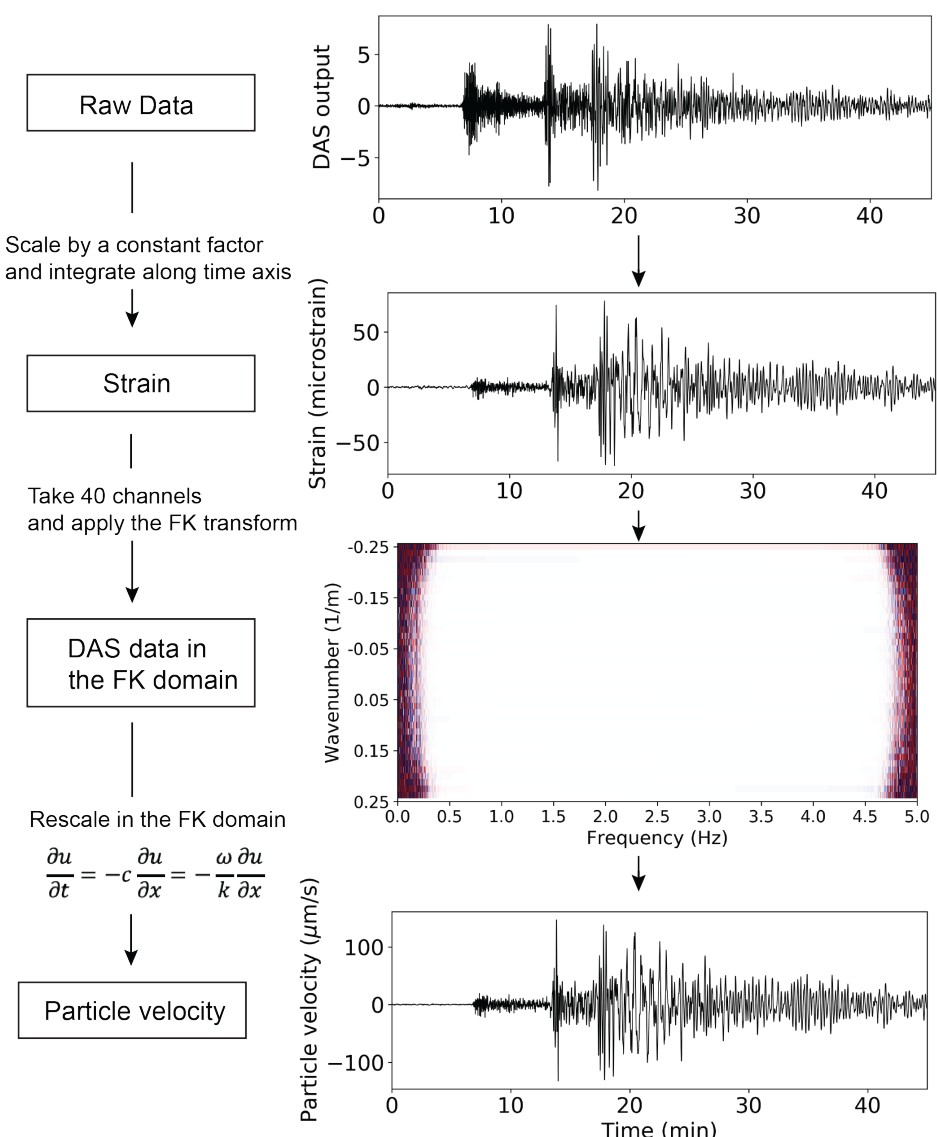

**Figure 3.** Left: Workflow of converting DAS measurements to particle velocity; right: an example of Peru M8.0 earthquake to demonstrate the workflow step by step.



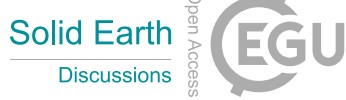

(a)

(b)

(c)

**Figure 4.** (a) Locations of 80 channels of the fiber array we used for calibration. Top shows the azimuth of two horizontal channels of SSPA, nearby seismometer 17.7 km away. (b) Comparison between channel 135 (black) with BH2 component of SSPA (red). (b) Comparison between channel 190 (black) with BH1 component of SSPA (red).


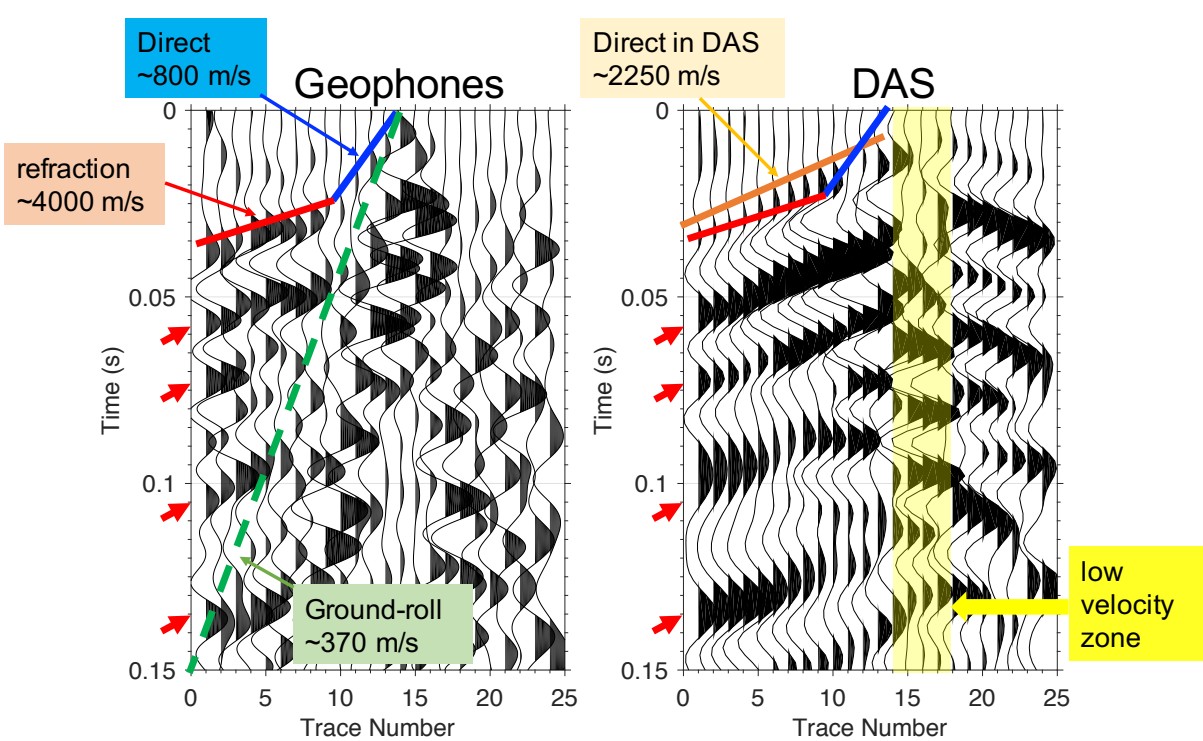

**Figure 5.** Geophone (left) and DAS (right) active-source shot gather.





**Figure 6.** DAS recordings of earthquakes: (a) Peru M8.0 on May 26 2019; (b) Ridgecrest M7.1 on July 06 2019; (c) Tennessee M3.8 on January 20 2020, and (d) PA M1.1 on August 27 2019. The top panel shows reference seismograms from nearby PSRS seismic station.





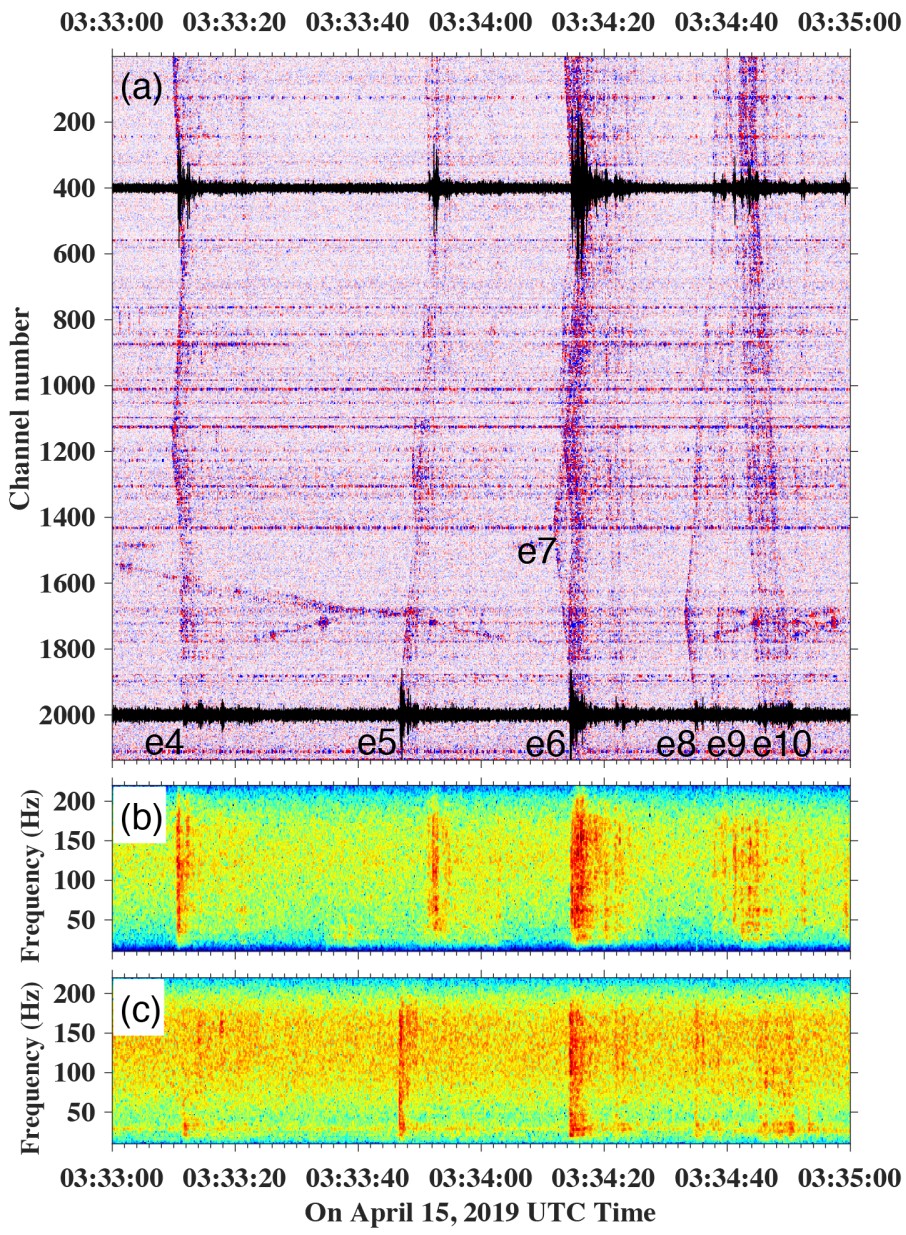

**Figure 7.** DAS recordings of (a) thunderquake events on April 15 2019.

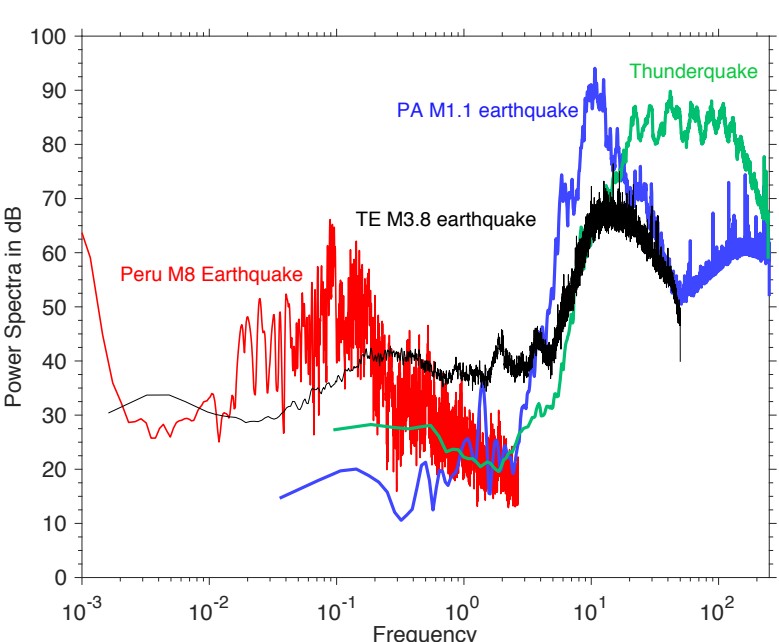

**Figure 8.** Power spectrum of DAS recordings of earthquakes and thunderquakes.





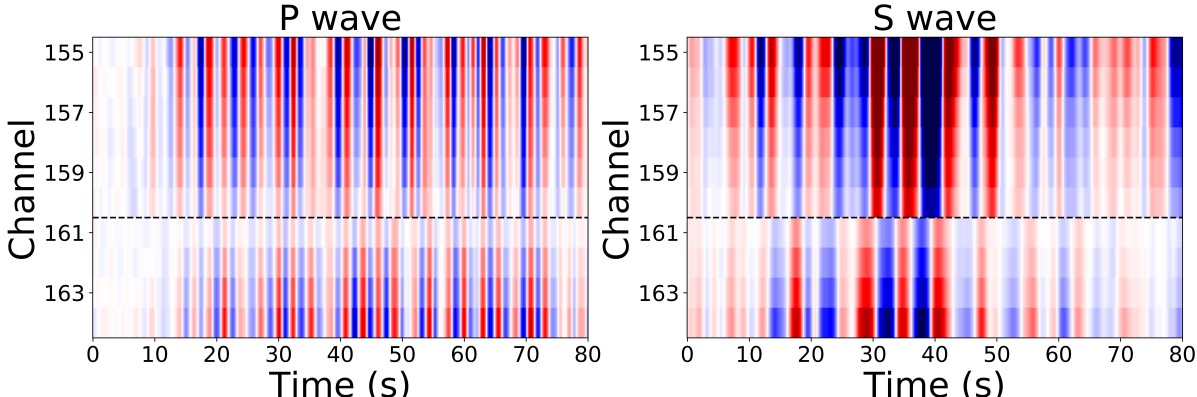

**Figure 9.** P wave and S wave traces observed in two orthogonal fibers (marked by dashed line in Figure 4a). There is a polarity flip of S wave.

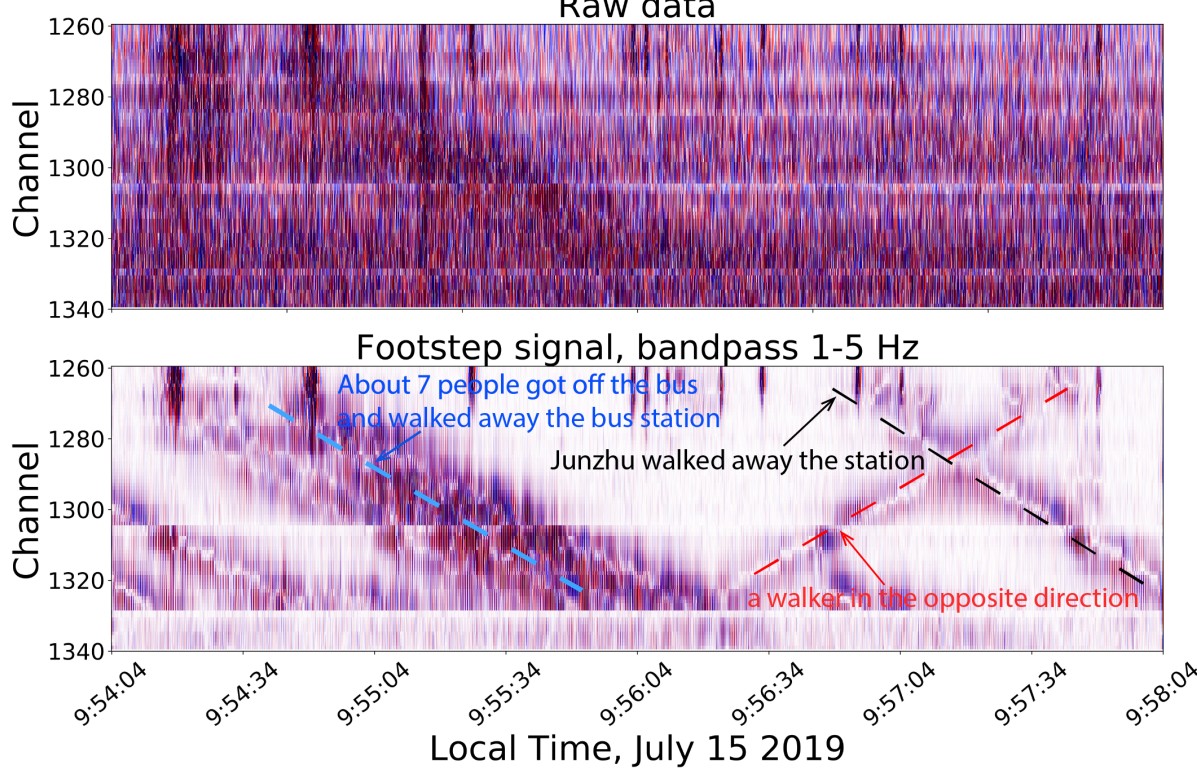

**Figure 10.** Top: 3-minutes raw data including footsteps; bottom: bandpassed data between 1 and 5 Hz. Black and blue dashed lines indicate a walker (author) and a group of walkers off the school bus walking away from the bus station (around channel 1270) along the fiber path, and later a walker moved towards the station. The slop of lines gives the walking speed of about 1.2 m/s.





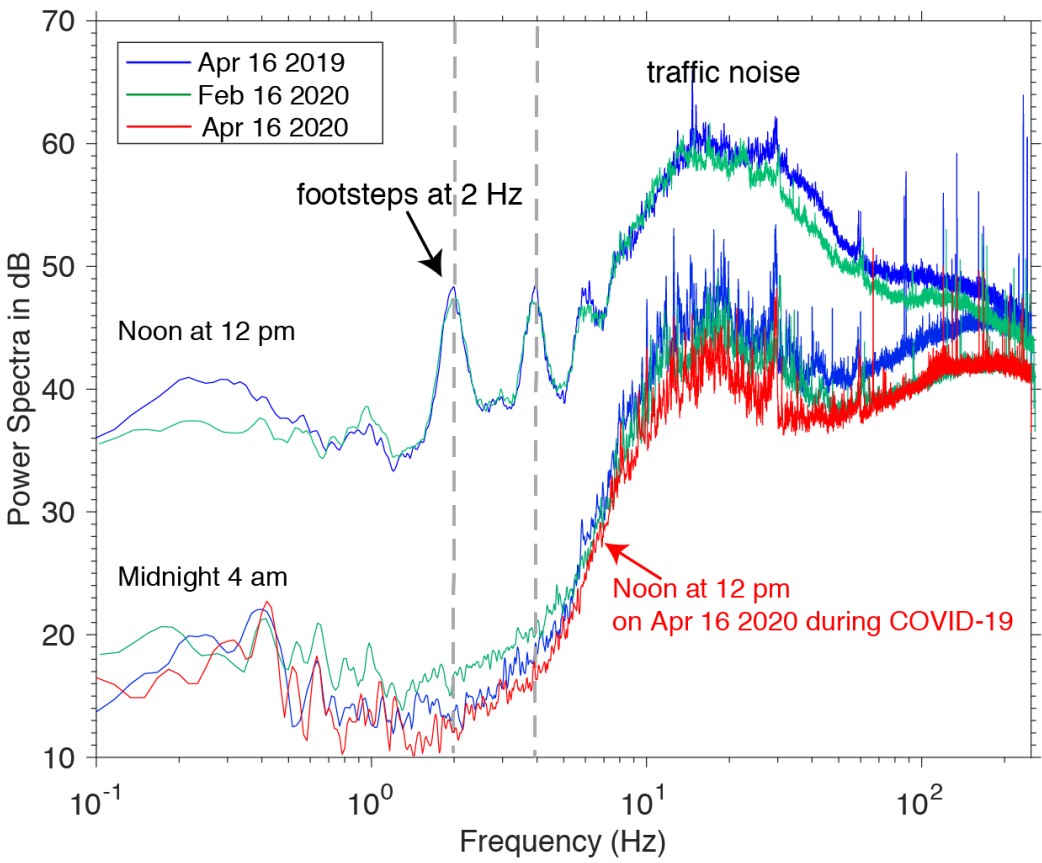

**Figure 11.** The power spectra of one-minute DAS data at three days (April 16 2019, February 16 2020, and April 16 2020) show that signals at noon are much stronger than signals at 4 am (UTC time). Additionally there are distinct peaks that appear in the signals from mid-day, particularly strong at 2 Hz and 4 Hz. Traffic noises correspond to > 5 Hz signals. During the COVID-19 pandemic the power spectra at noon is low as that at night time and no footstep signals are found.





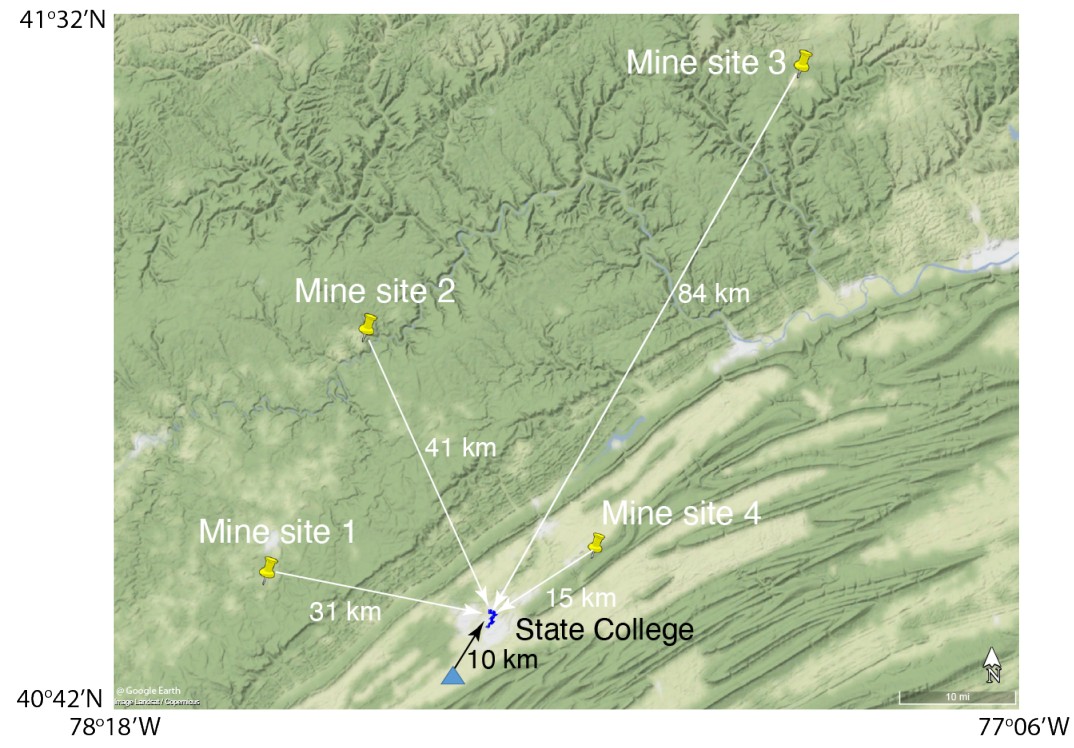

**Figure 12.** Map of geographical distribution of blast sites (site 1, site 2, site 3 and site 4) around the State College. The distances to State College are about 31 km, 41 km, 84 km, and 15 km, respectively. Seismic station indicated by the blue triangle is about 10 km west-south away from State College. Short blue line is the FORESEE array.





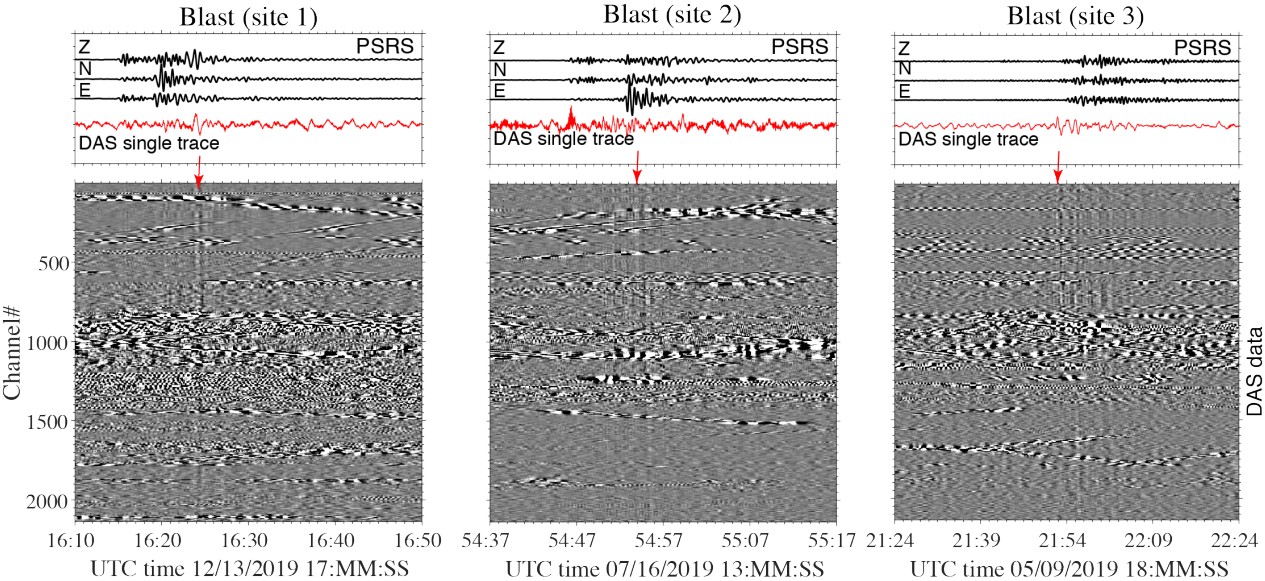

**Figure 13.** Seismic station (PSRS) and DAS recordings of three quarry blast signals from 3 mining sites (site 1, site 2, and site 3). Their magnitudes from Pennsylvania seismic network catalog are M1.3, M1.7, and M2.0. Red arrows indicate strong surface wave arrivals.



**Figure 14.** Seismic station (PSRS) and DAS recordings of one blast (M1.0) from site 4 (see 12), about 10 km away from State College. (a) three-component seismogram (bandpass filtering [1-15] Hz) and 100th DAS trace with two different bandpass filtering. DAS recording after (b) bandpass filtering [10-20] Hz and (c) bandpass filtering [1-5] Hz.

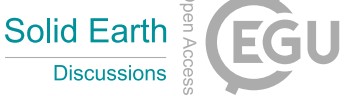

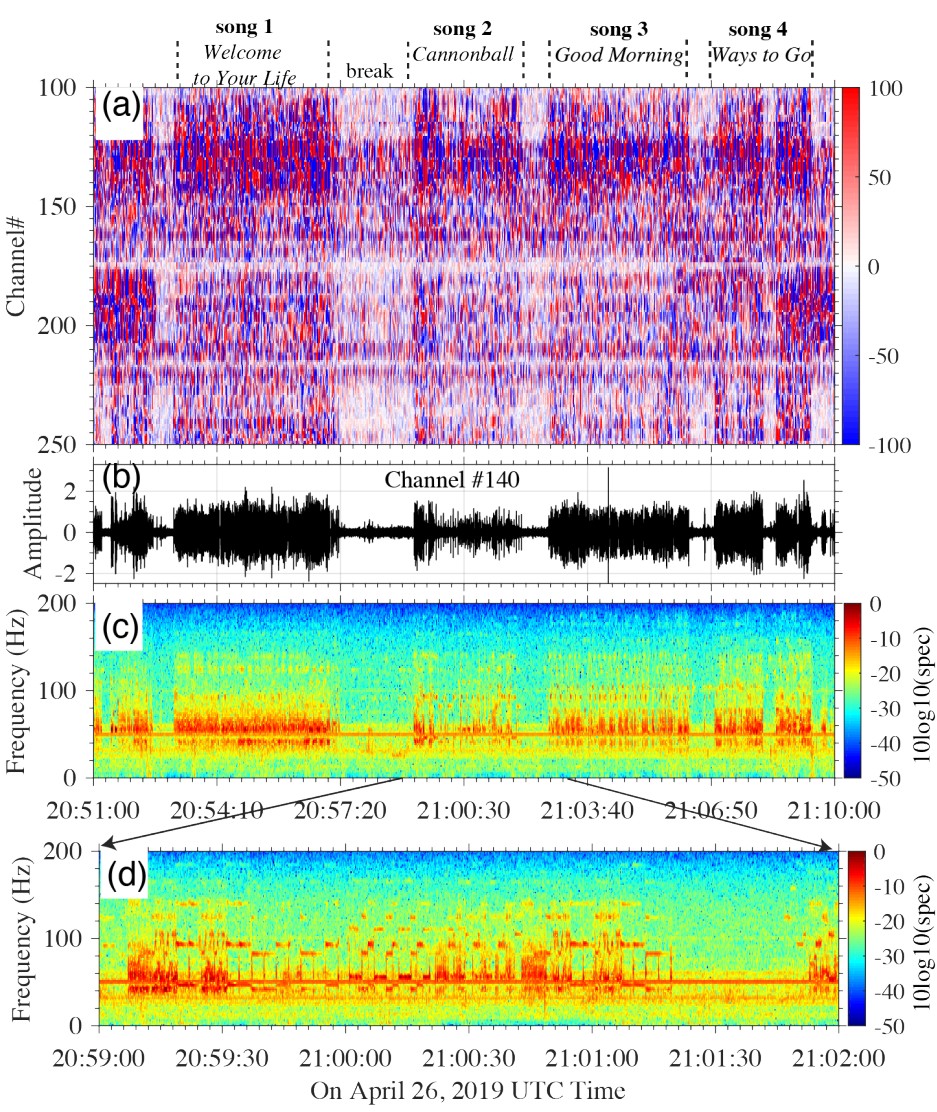

**Figure 15.** (a) DAS recording of concert music after bandpass filtering [1-150] Hz . (b) One trace at DAS channel 140, and (c) its spectrogram.

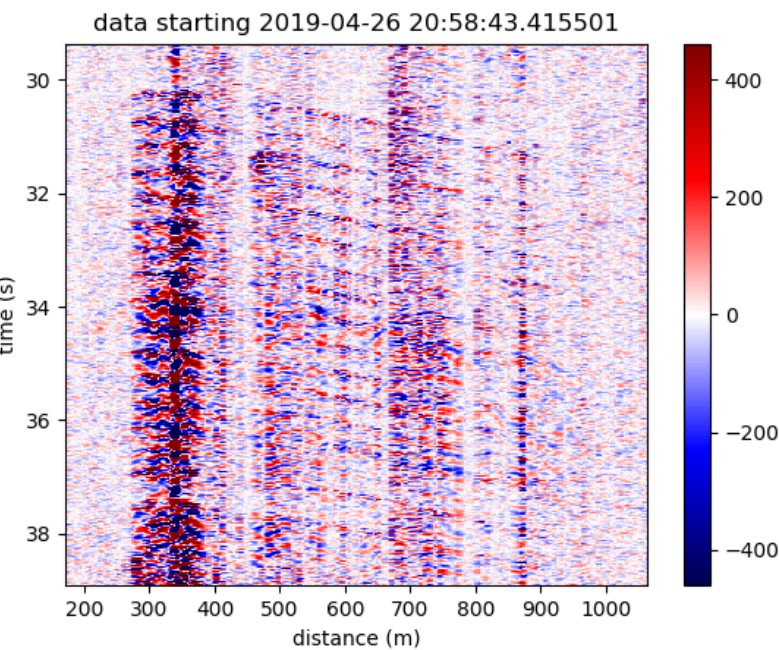

**Figure 16.** A short segment of seismic data from the beginning of song 2 in Figure 15 shows evenly spaced vibrations propagating away from the stage and detected over half a kilometer away.