# Peer review of "Sensing earth and environment dynamics by telecommunication fiber-optic sensors: An urban experiment in Pennsylvania USA"

_Solid Earth, 2020_

## Referee Comment (RC1) · Baoshan Wang (Referee) · 14 Jul 2020

In this paper, the authors presented the a one-year continuous DAS experiment, which is the first long term urban DAS practice with "dark fibre" in eastern US. The DAS records are callibarted with seismic records from a 3-component seismic station. The DAS recorded signals from natural earthquakes, thunderstorms, mining explosion, pace steps, and even live music. The experiment is interesting and the results are important references for future DAS applications. The paper is well written with appropriate analysises. Thus I would recommend publication after a minor revision with further clarification on following points.

[Figure]

Major points: 1, Two seismometers (SSPA, PSRS) were used respectively for calibration and signal comparison. If possible I would suggest to use the same station for different purpose. And please mark the relative location of different observation in at least one of the maps. 2, Line 157, "DAS surface energy is much stronger than S-wave", which is not obvious to me. Please further clarify this. 3, For the blast signal (section 5.2), I would expect the Rayleigh surface wave to be dominate, which is unlikely to show flipped polarity. Please further discuss this issue.

Minor points: 1, Please clarify the frequency band of seismometers used. 2, Please mark the arrival times of P, S, and surface waves in corresponding seismograms (Figs. 3,4,6,13,14). 3, Fig. 7, the subfigures are not capationed. 3, In fig. 8, please also provide the spectra of different signals from seismometer for better comparison. 4, Fig. 8, add the unit of horizontal axis (Hz).

---

## Referee Comment (RC2) · Anonymous Referee #2 · 29 Sep 2020

The paper shows an application of DAS measurement to permanent 2D network using dark fibers located in Pennsylvania Univ. State campus. The paper does not show really new results using DAS recordings, but it demonstrates quite synthetically how to turn a campus optical fiber network into a seismic network, its design and its sensitivity. This paper can thus serves as a reference for people interested in doing the same kind of experiment. However, in order to do so, the authors should reinforce the signal processing part with more detailed explanations, some demonstrations need to be clarified and they need to present figures in a way they can be used for comparison. Details are below.
Questions/remarks: 1) On line 130 (and figure 3) you explain your process to go from strain rate to velocity. At first sight, this operation should only require a single spatial integration. Can you explain in detail why you have to go through one time integration followed by one time derivation in addition to the 1/k integration? Could the process being done directly in the spatial Fourier domain using a high-pass wavenumber filter plus 1/k integration?

2) Line 139 mentions many factors that may influence waveform discrepancies. The authors mention the gauge length effect; it seems to me that the length of integration  $(40^{*}2=80m)$  compared to much larger wavelengths may also play a role?. It would be interesting to give more details and to show a more qualitative comparison (e.g. coherency), or to cite references that analyze this in detail. Nevertheless, it is good to see that the scaling factor is correct.

3) The lower frequency band specified on line 202 is a bit optimistic. According to figure 8, there is not much energy available below 0.02 Hz to bring such conclusion.

4) Figures 8 and 11 show power spectra with arbitrary unit. Please specify what is the reference for the dB scale you show.

5) What is the interest of figure 1b on lower right (fiber end)? Is it very small and we cannot deduce any information from it.

typos: - Is reference Martin et al, 2019, line 54, page 2 correct? This paper doesn't seem to deal with Stanford array - missing word at end of line 103

---

## Editor Comment (EC1) · Gilda Currenti (Editor) · 8 Oct 2020

Most important points to be addressed:

1) In Figure 3 the fk plot is illustrated. Please check the label over the x-axis. The plot seems symmetric, as it has been shown for negative and positive frequencies.

2) Always in Fig. 3, the strain is +- 50 microstrain, while particle velocity, derived for fk scaling, is +- 100 micro_m/s. This means that the average apparent velocity is 2 m/s. This outcome is not convincing. Please, check the results of the fk transform.

3) It is surprising to see a large strain variation of the order of microstrain for a tele-

seism. Estimates of dynamic strains at regional and teleseismic distances are available in literature. Please, refer to Agnew and Wyatt (2014). I suggest you to check carefully your conversion factor or the scale unit.

Minor points:

1) Fig. 4 correct the label in the y-axis from "partical" in "particle"

2) It would be helpful to have the colorbar scale in all the figures

---

## Author Comment (AC1) · 8 Oct 2020

Dear editor & reviewers,

We greatly appreciate your constructive comments to improve this paper, and we have revised our manuscript accordingly. According to all comments, we highlight all modifications in yellow in the revised manuscript. Below you will find point-by-point responses to these comments in the color blue.

**Reviewer #1** (by Dr. Baoshan Wang)
In this paper, the authors presented the a one-year continuous DAS experiment, which is the first long term urban DAS practice with "dark fibre" in eastern US. The DAS records are callibarted with seismic records from a 3-component seismic station. The DAS recorded signals from natural earthquakes, thunderstorms, mining explosion, pace steps, and even live music. The experiment is interesting and the results are important references for future DAS applications. The paper is well written with appropriate analysises. Thus I would recommend publication after a minor revision with further clarification on following points.
Thank you for positive recommendation. We addressed all your comments here below.

Major points:
Comment 1, Two seismometers (SSPA, PSRS) were used respectively for calibration and signal comparison. If possible I would suggest to use the same station for different purpose. And please mark the relative location of different observation in at least one of the maps.
We redrew Figure 4 with labels of the relative location of two stations SSPA and PSRS.

Comment 2, Line 157, "DAS surface energy is much stronger than S-wave", which is not obvious to me. Please further clarify this.
It is in Line 175. This sentence has been revised to "*Surprisingly, DAS has strong S-waves and seismometer surface wave energy is much stronger than S-wave.*". These phases are labeled in the new version of Figure 6.

Comment 3, For the blast signal (section 5.2), I would expect the Rayleigh surface wave to be dominate, which is unlikely to show flipped polarity. Please further discuss this issue.
We think that the strong energy is likely S-waves that exhibits the reversed polarity not surface wave. In section 5.2 (Line 259-261), we've rewritten the sentence as follow:
*Figure 14a shows high frequency P- and S-waves (10-20 Hz). In Figure 14b, we can see strong low frequency transverse motions. Same as previous observations, this low frequency transverse waves exhibits the flipped polarity in the orthogonal fiber locations (e.g., indicated by arrows in channels 170 and 600), which is either SH or Love wave and was also observed in previous DAS recordings from the Stanford DAS array (Martin, 2018; Fang et al., 2020).*

Minor points:
Comment 1, Please clarify the frequency band of seismometers used.
There was one missing statement of the bandpass frequency band in section 5.2. They are [1-2.5] Hz, which has been added in section 5.2.

Comment 2, Please mark the arrival times of P, S, and surface waves in corresponding seismograms (Figs. 3,4,6,13,14).
We've added phases in Figs 3, 4, 6, 13, 14.

Comment 3, Fig. 7, the subfigures are not capationed.
The caption (b) and (c) has been added in Figure 7.

Comment 4, In fig. 8, please also provide the spectra of different signals from seismometer for better comparison.
Since these DAS data represent strain rate, the power spectra here is the averaged value of all 2137 traces (channels). We think that detailed direct comparison is not informative in this case due to the distance between the DAS array and seismometer, differing near-surface conditions and different noise environment near the DAS array and seismometer. We would refer readers to Lindsey et al. (2020) JGR paper for full comparisons between a collocated seismometer and DAS. Below is the power spectra plot of all these signals for a nearby seismometer PSRS for reference.

[Figure]

Figure R1: Power spectra density of particle velocity records of four quakes recorded by nearby seismometer PSRS. Its unit of dB is relative to 1 (nano m/s)$^2$ /Hz.

*Reference: Lindsey, N. J., Rademacher, H., and Ajo-Franklin, J. B.: On the broadband instrument response of fiber-optic DAS arrays, Journal of Geophysical Research: Solid Earth, 125, e2019JB018 145, 2020.*

Comment 5, Fig. 8, add the unit of horizontal axis (Hz).
The unit "Hz" has been added in Figure 8.

**Reviewer #2**
The paper shows an application of DAS measurement to permanent 2D network using dark fibers located in Pennsylvania Univ. State campus. The paper does not show really new results

using DAS recordings, but it demonstrates quite synthetically how to turn a campus optical fiber network into a seismic network, its design and its sensitivity. This paper can thus serves as a reference for people interested in doing the same kind of experiment. However, in order to do so, the authors should reinforce the signal processing part with more detailed explanations, some demonstrations need to be clarified and they need to present figures in a way they can be used for comparison. Details are below.

Thank you for your positive comments. This paper aims to document the details of the new DAS array from installation to data recordings. In addition to earthquakes and mining blasts, this experiment reported several new DAS recordings for the first time, including the footstep signals, thunderquakes, and music signals. We expect that adding examples of these unique signals to the literature will have value for other groups struggling to identify the wide variety of signals detected by DAS arrays in populated areas. These new recordings extended previous knowledge of the surprisingly high sensitivity of DAS using underground telecom fibers. Potentially, DAS arrays could be useful tools for solving several geologic and environmental problems in Eastern US. New discovered sources (thunderquakes) may be potentially used for earth imaging in this region which has limited seismicity.

Below we addressed all your detailed questions.

Questions/remarks:
1) On line 130 (and figure 3) you explain your process to go from strain rate to velocity. At first sight, this operation should only require a single spatial integration. Can you explain in detail why you have to go through one time integration followed by one time derivation in addition to the 1/k integration? Could the process being done directly in the spatial Fourier domain using a high-pass wavenumber filter plus 1/k integration?

Figure 3 shows individual step to convert strain rate $\frac{\partial \varepsilon}{\partial t}$ to velocity $v$:

1) Strain rate $\frac{\partial \varepsilon}{\partial t}$ to strain $\varepsilon$ by time integration
2) Strain $\varepsilon$ in the FK domain
3) Rescale strain by $v(f,k) = -\frac{\omega}{k} \varepsilon(f,k)$
4) Apply inverse Fourier transform to $v(f,k)$ to get velocity $v$

Along with each step, we show the data in the right side. We refer readers to other references (e.g., Daley et al., 2016; Wang et al., 2018) for different conversion methods.

*Reference: Daley, T., Miller, D., Dodds, K., Cook, P., and Freifeld, B.: Field testing of modular borehole monitoring with simultaneous distributed acoustic sensing and geophone vertical seismic profiles at Citronelle, Alabama, Geophysical Prospecting, 64, 1318–1334, 2016.*
*Wang, H. F., Zeng, X., Miller, D. E., Fratta, D., Feigl, K. L., Thurber, C. H., and Mellors, R. J.: Ground motion response to an ML 4.3 earthquake using co-located distributed acoustic sensing and seismometer arrays, Geophysical Journal International, 213, 2020–2036, 2018.*

2) Line 139 mentions many factors that may influence waveform discrepancies. The authors mention the gauge length effect; it seems to me that the length of integration (40*2=80m) compared to much larger wavelengths may also play a role?. It would be interesting to give more details and to show a more qualitative comparison (e.g. coherency), or to cite references that analyze this in detail. Nevertheless, it is good to see that the scaling factor is correct.

In our calibration following Wang et al., 2018, we use f-k transform to calculate the scale factor $\frac{\omega}{k} = c$ to complete the conversion $v(f,k) = -\frac{\omega}{k}\varepsilon(f,k)$. This 80 meter length (40 channels for clean signals) allows us to grab a collection of nearby channels to yield a 2D waveform (x,t) which is required to calculate the f-k transform (x,t)→ (k,f). We repeat our data processing flow with longer length 800 m. Due to large waveform variation across channels (Fig s2a) possibly caused by DAS instrument response, different scaling factors resulted in different waveforms (Fig s2b). This waveform difference from many channels may mix with DAS instrument response slightly (Lindsey et al. 2020), who calibrated the DAS instrument response using the ratio of seismometer trace and DAS trace. The final result of this process is a velocity equivalent trace for each channel (not an average/stack of the channels).

[Figure]

Figure s2: (a) waveform in Channel 170 – 590; (b) scaled waveform with different length against seismometer waveform.

In the manuscript, we would refer to Wang et al. 2018 paper and Lindsey et al. 2020 for details of unit conversion since they have full data comparison with collocated seismometers and DAS.

*Reference: Wang, H. F., Zeng, X., Miller, D. E., Fratta, D., Feigl, K. L., Thurber, C. H., and Mellors, R. J.: Ground motion response to an ML 4.3 earthquake using co-located distributed acoustic sensing and seismometer arrays, Geophysical Journal International, 213, 2020–2036, 2018.*
*Lindsey, N. J., Rademacher, H., and Ajo-Franklin, J. B.: On the broadband instrument response of fiber-optic DAS arrays, Journal of Geophysical Research: Solid Earth, 125, e2019JB018 145, 2020.*

3) The lower frequency band specified on line 202 is a bit optimistic. According to figure 8, there is not much energy available below 0.02 Hz to bring such conclusion.
Since the frequency content of the teleseismic recording of the Peru earthquake shows strong low-end frequency response (red curve in Figure 8 with uptick in energy on the left side), this leads us to see that lower frequencies can be recorded by DAS as low as 0.001 Hz.

4) Figures 8 and 11 show power spectra with arbitrary unit. Please specify what is the reference for the dB scale you show.
Figures 8 & 11 show the averaged power spectra density of the strain rate DAS trace in dB with respect to 1 $(nanostrain/sec)^2$ /Hz. We average the PSD (in dB) across 2137 channels from the DAS array. So the output is the average PSD of all 2137 traces.

5) What is the interest of figure 1b on lower right (fiber end)? Is it very small and we cannot deduce any information from it.
This photo shows a particular way to terminate the fiber by wrapping around the pencil to reduce the noise. Some other DAS interrogator units require the use of specialized termination hardware, or require that the fiber makes a loop back to the interrogator, but the one we use can be simply terminated in this way.

typos: - Is reference Martin et al, 2019, line 54, page 2 correct? This paper doesn't seem to deal with Stanford array
It's a typo and we refer to Martin's PhD thesis (2018).

- missing word at end of line 103
This sentence is fixed in the revised manuscript.

We hope the responses above address your comments and answer your questions satisfactorily. Thank you very much for your review and we truly appreciate your comments.

Sincerely,

Tieyuan Zhu
Penn State Geosciences
On the behalf of co-authors.

---

## Author Comment (AC3) · 13 Oct 2020

Dear Gilda,

Thank you for careful checking. Please find the point-to-point response to all your comments below:

1) In Figure 3 the fk plot is illustrated. Please check the label over the x-axis. The plot seems symmetric, as it has been shown for negative and positive frequencies.

Reply: you're right. This fk plot was incorrectly shown here and reversed x and y axis. We have updated figure 3 in the revised manuscript (see attached figure 3 as well).

[Figure]

2) Always in Fig. 3, the strain is +- 50 microstrain, while particle velocity, derived for fk scaling, is +- 100 micro_m/s. This means that the average apparent velocity is 2 m/s. This outcome is not convincing. Please, check the results of the fk transform.

Reply: This unit 'microstrain' is a typo. The unit for strain (2nd figure in Fig 3) is nanostrain.

3) It is surprising to see a large strain variation of the order of microstrain for a tele-seism. Estimates of dynamic strains at regional and teleseismic distances are available in literature. Please, refer to Agnew and Wyatt (2014). I suggest you to check carefully your conversion factor or the scale unit.

Reply: The strain of this teleseismic is nanostrain from DAS.

Minor points: 1) Fig. 4 correct the label in the y-axis from "partical" in "particle"

Reply: This typo in the label of Fig. 4 has been corrected in the revised manuscript.

2) It would be helpful to have the colorbar scale in all the figures

Reply: The colorbar has been added for all the figures.

Thank you, Tieyuan Zhu

———————————————

Raw strain rate data

Scale by a constant factor
and integrate along time axis

Strain

Take 40 channels
and apply the f-k transform

Strain in the f-k
domain

Rescale strain by $-\frac{\omega}{k}$

Particle velocity

**Fig. 1.** figure 3

---

## Editor Comment (EC2) · Gilda Currenti (Editor) · 16 Oct 2020

1) The argumentations raised by the comment 1 and 2 of the second reviewer need to be included in the manuscript.

2) Lindsey 2020 shows that the conversion from strain to velocity using the fk rescaling method is strongly dependent on the threshold parameter. Which value have you used? Did you find a similar sensitivity?

3) You have reported well how to calibrate the DAS records using the scaling factor to obtain signal in strain rate unit. It is important to estimate the amplitude of the strain

rate signals, especially for the new class of records that you show for the first time in the manuscript. So, please convert the figures in the manuscript from DAS unit to strain rate.

---

## Author Comment (AC4) · 19 Oct 2020

1) The argumentations raised by the comment 1 and 2 of the second reviewer need to be included in the manuscript.

Reply: We highlighted line 130-140 for comment 1. We've added a new paragraph in the manuscript to response comment 2 (Line 140-146). Thank you.

2) Lindsey 2020 shows that the conversion from strain to velocity using the fk rescaling method is strongly dependent on the threshold parameter. Which value have you used? Did you find a similar sensitivity?

[Figure]

Reply: We used 40 traces with the sigma (1e-5) and 400 traces with 1e-3. It is not surprising to consider highly variable waveform within 400 traces. Another factor to consider is the unknown DAS instrument response. We refer readers to Lindsey et al. 2020 and further discussion of this conversion is beyond of the scope of this article.

Lindsey, N. J., Rademacher, H., and Ajo-Franklin, J. B.: On the broadband instrument response of fiber-optic DAS arrays, Journal of Geophysical Research: Solid Earth, 125, e2019JB018 145, 2020.

3) You have reported well how to calibrate the DAS records using the scaling factor to obtain signal in strain rate unit. It is important to estimate the amplitude of the strain rate signals, especially for the new class of records that you show for the first time in the manuscript. So, please convert the figures in the manuscript from DAS unit to strain rate.

Reply: Thanks for this suggestion. All figures of DAS data are plotted in strain rate in the revised manuscript.

---

## Referee Report (RR1)

This review follows my first comments sent during the discussion process.
I still have two questions for which the answers given by the authors do not satisfy me.

1)
The strain-rate to velocity processing includes a time integration that requires a highpass filtering to get rid of low frequency noise, and that is ended by a –w/k multiplication that I understand as being simultaneous time derivation and wavenumber integration.
Why is it necessary to first integrate w/r to time and then differentiate w/r to time? Could this time integration/derivation be avoided?

The answer given by the authors to my same remark was to refer to Daley et al., 2016; Wang et al., 2018 for different conversion methods.

Daley et al. 2016 use borehole data and assume body wave propagating along the vertical profile at a constant, non dispersive speed of 3500m/s.
In that case, it makes sense to transform the w/k multiplication by a simple scaling w/k=c=3500m/s. The whole process requires only one time integration and additional scaling.
In the FORESEE experiment, waves can be various, dispersive or non dispersive, and I don't see any way to infer characteristic c values.

Wang et al, 2018 use a similar approach to the present paper, first integrating w/r to time, then derivating w/r to time.

These two papers don't explain me why this time integration/derivation step is needed. The issues are i) in signal processing, integration steps are avoided when possible ii) The authors choose to detail the data processing, this is a good idea but it requires making things clear to the reader.
To conclude:
- Do I miss something in the w/k scaling step that is not explained (apparent velocities are estimated first)?
- Is this processing just applied because you are used to it, but there are no real justifications for it except that strain rate integration is performed automatically, or it comes as a standard toolbox? This would be an acceptable reason with the proper explanation.

2) I still disagree with the authors' statement concerning the frequency range specified on line 206. I try to illustrate it from snapshots taken from the paper.
There is no PSD relative to the noise level of the instrument or comparison with the earth noise level at very low frequency. This would have been interesting though. Figure 11 give us some numbers for this noise down to 0.1 Hz

[Figure]

It varies from 15-20db at night to 35-40 at noon. On figure 8, left part of the Psd, Peru M8 earthquake, we reach this noise level somewhere in-between 0.02 and 0.01Hz. According to the text, no filtering has been applied.

If we refer to the relationship between strain-rate and velocity, that is roughly: strain-rate $\sim$ w/c*velocity, we should observe a spectrum shape that is twice the derivative of the displacement spectrum for a Mag8 quake whose corner frequency is between 1e-3Hz and 1e-2 Hz.

This is to say that I don't see any reason why the source spectrum should increase below 0.002Hz. It should exhibit a bell shape with a maximum between 1mHz and 10 mHz

I think that the instrument/earth noise level is reached at $\sim$ 1.e-2Hz, and below the spectrum has nothing to do with the EQ. The DAS records quite well this earthquake down to 1.e-2Hz, that is already remarkable.

---

## Author Response (AR2)

Dear editor and reviewers,

Thank you again for considering our manuscript as well as providing detailed explanation on the DAS unit conversion and DAS noise level.

Please find the point-to-point response below.

1. Why is it necessary to first integrate w/r to time and then differentiate w/r to time? Could this time integration/derivation be avoided?
   No, please follow our Figure 3 workflow. Vel = -w/k*Strain-rate

2. - Do I miss something in the w/k scaling step that is not explained (apparent velocities are estimated first)?
   w/k aims to fine the optimal apparent velocity c. This has been explicitly explained in Line 140.

3. - Is this processing just applied because you are used to it, but there are no real justifications for it except that strain rate integration is performed automatically, or it comes as a standard toolbox? This would be an acceptable reason with the proper explanation.
   We used this processing. We realized that the waveform matching is somehow also dependent on the threshold number \sigma and the apparent velocity c for Peru M8 earthquake plane wave. The velocity is likely ambiguous with limited offset that is the case of the FORESEE array. We've added additional comments in Line 140-146 in the revised manuscript. We hope it is clear for readers when they will conduct the unit conversion.

4. I think that the instrument/earth noise level is reached at ~ 1.e-2Hz, and below the spectrum has nothing to do with the EQ. The DAS records quite well this earthquake down to 1.e-2Hz, that is already remarkable.
   Thank you for detailed explanation of this frequency. We agreed with you on this. This has been changed to 1e-2 Hz in Line 216 in the revised manuscript.

We hope the responses above address your comments and answer your questions satisfactorily. Thank you very much for your review and we truly appreciate your comments.

Sincerely,

Tieyuan Zhu
Penn State Geosciences
On the behalf of co-authors.

[revised manuscript text omitted]

---

## Author Response (AR3)

Dear editor Dr. Currenti,

Dear Authors,
I found the last version of the manuscript improved. There is still just a minor revision that I suggest
before pubblication. At line 140 you mention that "The f-k transform aims to find the scaling factor
velocity c." Indeed, the method is used to convert and compare strain and velocity data. In such a
case, the velocity c is not just a scaling factor. I suggest to remove this sentence.
Best Regards
Gilda Currenti

As suggested, we have removed the sentence in Line 140 in the revised manuscript.

Thank you for your time,

Tieyuan Zhu
On behalf of authors

[revised manuscript text omitted]